# SPI-1 virulence gene expression modulates motility of *Salmonella* Typhimurium in a proton motive force- and adhesins-dependent manner

Doaa Osama Saleh[1,2]*, Julia A. Horstmann[3], María Giralt-Zúñiga[1], Willi Weber[4], Eugen Kaganovitch[5], Abilash Chakravarthy Durairaj[3,6], Enrico Klotzsch[4], Till Strowig[6,7], Marc Erhardt[1,8]*

1 Institute for Biology/Molecular Microbiology, Humboldt-Universität zu Berlin, Berlin, Germany, 2 Department of Microbiology and Immunology, Faculty of Pharmacy, Ain Shams University, Cairo, Egypt, 3 Junior Research Group Infection Biology of Salmonella, Helmholtz Centre for Infection Research, Braunschweig, Germany, 4 Institute for Biology, Experimental Biophysics/Mechanobiology, Humboldt-Universität zu Berlin, Berlin, Germany, 5 Max Planck Institute for Terrestrial Microbiology and Center for Synthetic Microbiology (SYNMIKRO), Marburg, Germany, 6 Department of Microbial Immune Regulation, Helmholtz Centre for Infection Research, Braunschweig, Germany, 7 Cluster of Excellence RESIST (EXC 2155), Hannover Medical School, Hannover, Germany, 8 Max Planck Unit for the Science of Pathogens, Berlin, Germany

* doaa.saleh@hu-berlin.de (DOS); marc.erhardt@hu-berlin.de (ME)

**Data Availability Statement:** All data supporting the conclusions are available in the manuscript's main or supplementary files.

## Abstract

Both the bacterial flagellum and the evolutionary related injectisome encoded on the *Salmonella* pathogenicity island 1 (SPI-1) play crucial roles during the infection cycle of *Salmonella* species. The interplay of both is highlighted by the complex cross-regulation that includes transcriptional control of the flagellar master regulatory operon *flhDC* by HilD, the master regulator of SPI-1 gene expression. Contrary to the HilD-dependent activation of flagellar gene expression, we report here that activation of HilD resulted in a dramatic loss of motility, which was dependent on the presence of SPI-1. Single cell analyses revealed that HilD-activation triggers a SPI-1-dependent induction of the stringent response and a substantial decrease in proton motive force (PMF), while flagellation remains unaffected. We further found that HilD activation enhances the adhesion of *Salmonella* to epithelial cells. A transcriptome analysis revealed a simultaneous upregulation of several adhesin systems, which, when overproduced, phenocopied the HilD-induced motility defect. We propose a model where the SPI-1-dependent depletion of the PMF and the upregulation of adhesins upon HilD-activation enable flagellated *Salmonella* to rapidly modulate their motility during infection, thereby enabling efficient adhesion to host cells and delivery of effector proteins.

## Author summary

The injectisome encoded on *Salmonella* pathogenicity island-1 (SPI-1) and the bacterial flagellum are two important virulence factors that are required by *Salmonella* Typhimurium

**Funding:** This work was supported in part by a project that has received funding from the European Research Council (ERC) under the European Union's Horizon 2020 research and innovation programme (grant agreement n° 864971) and from the VolkswagenStiftung (to M. E.) and by a GERLS scholarship (n° 91705821) co-funded by the Egyptian Ministry of Higher Education and Scientific Research (MHESR) and the German Academic Exchange Service (Deutscher Akademischer Austauschdienst (DAAD)) (to D.O.S.). The funders had no role in study design, data collection and analysis, decision to publish, or preparation of the manuscript.

**Competing interests:** The authors have declared that no competing interests exist.

to establish an infection in the host's intestine. Previously, we had uncovered a regulatory cross-talk, where the SPI-1 regulator HilD activated expression of *flhDC*, the flagellar master regulator. However, the physiological consequences of this interaction were unclear. Here, we found that the activation of HilD surprisingly results in a significant reduction of motility, which was independent of HilD-mediated activation of *flhDC* expression. Our results further demonstrate that HilD expression results in the upregulation of several adhesive structures, activation of the stringent response and depletion of the proton motive force, which is required for energizing flagella rotation. From these findings, we propose a model where the depletion of proton motive force and upregulation of adhesins upon HilD activation allow flagellated *Salmonella* to quickly adjust their motility during the infection process. This rapid modulation of motility might facilitate efficient adhesion of the bacteria to the host cell surface and the delivery of effector proteins.

## Introduction

*Salmonella enterica* is a rod-shaped, Gram-negative enteropathogen that causes human salmonellosis, including gastroenteritis and in rare cases enteric fever [1]. Important virulence factors include the flagellum and the injectisome, which is used to deliver effector proteins into host cells [2]. The flagellum is a rotating motility organelle that enables directed swimming towards nutrients and away from harmful substances [3]. Moreover, flagella are thought to contribute to *Salmonella* pathogenesis by enhancing the interaction with the host cell membranes and promoting actin polymerization [4,5]. Flagella are made of three main structures: (i) the basal body, embedded in the inner and outer membranes of *Salmonella*, (ii) a flexible hook, and (iii) a 10–15 μm long, helical filament [6]. The expression of flagella genes is regulated in a complex transcriptional hierarchy [7]. On top of the cascade resides the flagellar master operon *flhDC*. Although six transcriptional start sites have been mapped within the promoter region of *flhDC*, only P1 and P5 were proved to be functional in a wild type background, where they drive its expression during early and late growth phases, respectively [8,9]. The transcription of *flhDC* is under the control of a $\sigma^{70}$-dependent class I promoter and the protein products assemble in a heteromultimeric complex ($FlhD_4C_2$), which ultimately controls flagellar gene expression and flagellar assembly [9–11]. Functional $FlhD_4C_2$ directs the $\sigma^{70}$/RNA polymerase holoenzyme to initiate transcription from class II promoters [12]. This results in the production of proteins needed for the construction of the basal body and hook, as well as regulatory proteins, such as the flagellar sigma factor FliA ($\sigma^{28}$). FliA directs RNA polymerase to transcribe flagellar genes under the control of class III promoters. The products of those genes constitute the filament, the flagellar motor and chemosensory systems [13,14].

Injectisomes are specialized syringe-like needle complexes employed by *Salmonella* to inject effector proteins into eukaryotic host cells, thus manipulating them and facilitating the establishment of a successful infection [15,16]. The injectisome systems of *Salmonella* are encoded on the *Salmonella* Pathogenicity Islands 1 (SPI-1) and 2 (SPI-2) and play a role at different stages of the infection. The SPI-1-encoded injectisome is involved in the initial step of infection, where it translocates effector proteins into cells of the intestinal epithelium resulting in actin polymerization and remodelling [17,18]. This cytoskeletal remodelling results in ruffle formation on the host cell surface promoting the internalization of the bacterial cells. The SPI-2-encoded injectisome is crucial in the following steps, where it promotes trafficking of *Salmonella* across the epithelial cells as well as their replication and survival inside macrophages [19,20].

A battery of DNA-binding proteins, such as the AraC-like transcriptional regulators HilD, HilC and RtsA, regulate expression of SPI-1 genes. Each of those regulators can activate in a feed-forward loop its own transcription and the transcription of the other two regulators [21,22]. The three regulators can independently activate the expression of the transcription factor HilA, which harbors a OmpR/ToxR-type DNA binding domain and functions as the SPI-1 master regulator [23]. However, HilD plays the major role in the transcriptional activation of *hilA* in response to the environmental cues, while HilC and RtsA serve as amplifiers of this signal [22]. HilA then directly activates the expression of SPI-1 injectisome structural components by binding to the SPI-1 encoded *prg/org* and *inv/spa* operons [24,25]. Additionally, it activates the expression of *invF*, an AraC-like transcriptional activator, as well as the SPI-1 chaperone *sicA*. InvF then interacts with SicA to induce the expression of effector proteins [26]. On the other hand, the transcription of *hilD* is negatively regulated by the SPI- encoded regulator HilE, which results in transcriptional repression of *hilA* and the downstream SPI-1 genes [27].

Both the flagellum and the injectisome are evolutionary related and, in particular, share a homologous protein export system, termed type-III secretion system [28,29]. To synthesize, assemble and secrete components of both systems, as well as to ensure flagellar rotation, significant amounts of energy are required [30–32]. The secretion process in both systems is proton motive force (PMF)-dependent and coupled to ATP hydrolysis [33–36].

Moreover, *Salmonella* has the ability to rapidly adapt to multiple different conditions during the infection process. This requires a precise coordination of the production and operation of the injectisomes and the flagellum, which are both energy demanding processes. Several cross-regulatory connections between the different genetic systems that potentially contribute to this tight coordination process have been described in the literature (Fig 1A). Kage et al. showed that the flagellar protein FliZ, which post-translationally activates FlhDC, regulates HilD at the post-transcriptional level [37,38]. Their results indicate a decrease in HilD translation levels upon *fliZ* deletion with no considerable change in protein stability. This effect was more pronounced in cells deleted for the protease ClpXP, which acts as a negative regulator of flagellum biosynthesis via its proteolytic action on FlhDC and thus its deletion is expected to result in increased cellular levels of FliZ. In contrast, expression of the flagellar master operon *flhDC* is influenced by various transcription factors, such as RflM/RcsB [39] or RtsB, which is transcribed in an operon together with the SPI-1 regulator RtsA [40]. RtsB binds within the *flhDC* promoter region to repress flagella biosynthesis. Additionally, FlhDC is post-transcriptionally regulated by the anti-FlhDC factor RflP (formerly known as YdiV) in response to nutrient levels and cell envelope stress [41,42]. HilD is also known to be involved in other regulatory networks in addition to its well-established role as an activator of SPI-1 gene expression [43]. In most of the so-far characterized cases, it exerts its regulatory function via relieving the repressive effect of the histone-like nucleoid-structuring protein (H-NS) on the promoters of the regulated genes [44–46]. For example, HilD activates the expression of SPI-2 genes via antagonizing the H-NS repression on the promoter region of the SPI-2 encoded *ssrAB* operon, favouring its OmpR-dependent expression [44,47]. SsrAB activates in turn the expression of SPI-2 genes. The expression of SPI-4 genes, which encode a PMF-dependent type-1 secretion system (T1SS) and its substrate protein SiiE, were also shown to be co-regulated with SPI-1 [48,49]. Further, SPI-5 encodes effectors that are translocated by the SPI-1-T3SS, are negatively regulated by H-NS and co-activated with SPI-1 genes under the same environmental conditions, which suggests that HilD is involved in their activation process [50,51]. In a previous study, we additionally described a transcriptional cross talk between SPI-1 and flagellar genes, where HilD directly binds within the promoter region of *flhDC* upstream of the P5 transcriptional start site and activates its expression [52].

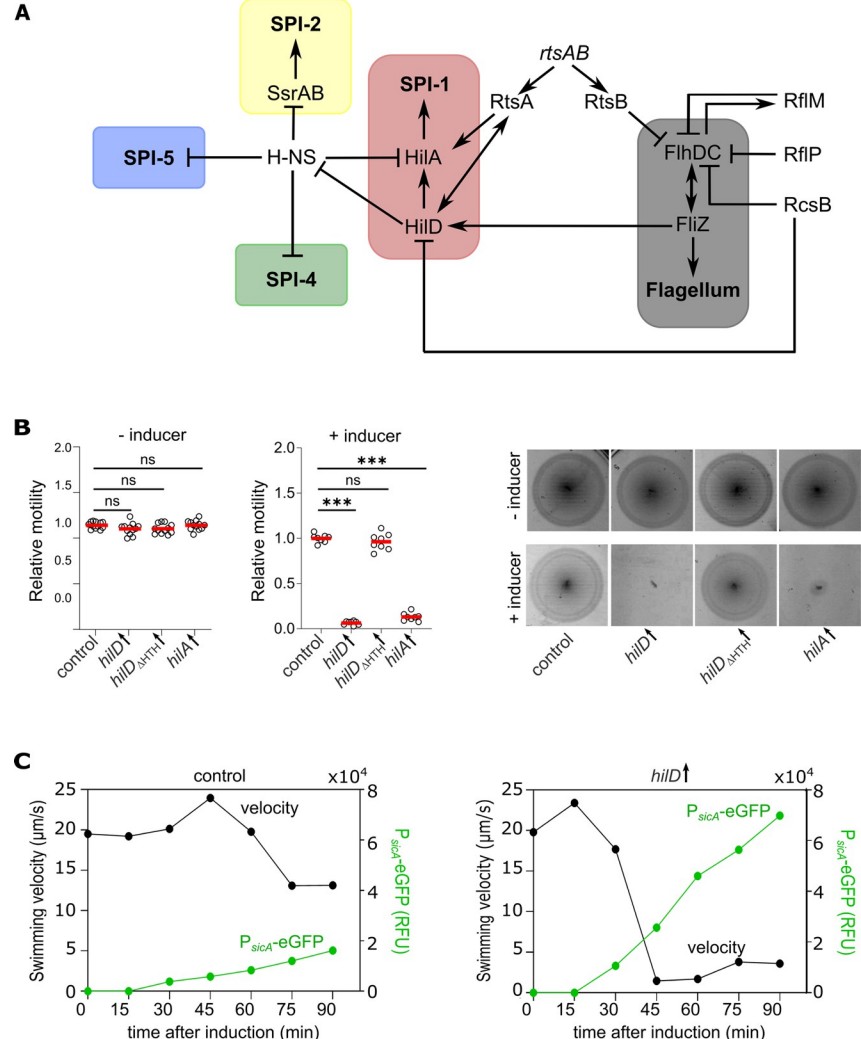

**Fig 1. Activation of HilD abrogates swimming motility.** (A) Cross-talk between different *Salmonella* regulons. Only relevant cross-regulations are shown. (B) Swimming motility in soft-agar swim plates was monitored at 37°C for 3.5 h in absence or presence of 0.2% arabinose to induce HilD production. Diameters of swimming halos were measured and normalized to the control (left and middle panels). Representative swimming halos of the analysed mutants are shown (right). Biological replicates are shown as individual data points. Horizontal bars (red) represent the calculated mean of at least eight biological replicates. Statistical significances were determined using a two-tailed Student's t-test (***, $P < 0.001$; ns, $P > 0.05$). Strains analysed were EM808 (control), TH16339 (*hilD*↑), EM831 (*hilD*$_{\Delta HTH}$↑) and EM930 (*hilA*↑). (C) Induction of SPI-1 gene expression was monitored using a P$_{sicA}$-eGFP transcriptional reporter fusion. Fluorescence intensities of P$_{sicA}$-eGFP fusion and single-cell swimming velocities were measured after induction of HilD production by addition of AnTc (100 ng/ml). Fluorescence intensities of the cultures were measured in a microplate reader and normalised to the measured OD$_{600}$ to give relative fluorescence units (RFU). The swimming velocities were analysed via time-lapse microscopy. The plotted values represent the average of 300 or more single-cell tracks from a representative experiment. Strains analysed were EM228 (control) and EM12302 (*hilD*↑). *hilD*↑: strain expressing *hilD* under an inducible promoter. *hilA*↑: strain expressing *hilA* under an inducible promoter. AnTc: anhydrotetracycline.

In the present study, we report the surprising finding that activation of chromosomal HilD expression results in a pronounced motility defect in contrast to the increased flagellar gene expression upon *flhDC* activation. Transcriptome analysis revealed the HilD-dependent upregulation of genes encoding various adhesive structures such as chaperone-usher fimbriae or

curli. Single cell analyses revealed that HilD-induced cells activated the stringent response and displayed a decrease in their membrane potential. Our results suggest that the observed motility defect upon HilD-activation is a multifaceted process that affects various cellular events. We propose that upregulation of adhesive structures and depletion of the PMF after activation of SPI-1 injectisome expression might allow *Salmonella* to rapidly adjust their motility behaviour during host cell infection independent of flagellar gene expression and assembly.

## Results

### Motility defect upon HilD-activation

We have previously shown that HilD directly activates flagellar gene expression by binding to the P5 transcriptional start site of the *flhDC* promoter [52]. However, the physiological relevance of the HilD-dependent activation of *flhDC* gene expression remained unclear. To characterize the HilD-dependent motility phenotype, we generated strains chromosomally expressing *hilD* under the control of an inducible promoter, either from an ectopic locus (*araBAD*), while retaining the native copy, or solely from the native locus where the native promoter has been replaced by a tetracycline-inducible promoter ($P_{tet}$). HilD-dependent activation of gene expression was verified by monitoring the transcriptional activity of the promoter of the SPI-1 gene *sicA* fused to eGFP ($P_{sicA}$-eGFP) and secretion of SPI-1-encoded effector proteins. As expected, transcription of *sicA* (S1A Fig) and secretion of effector proteins into the culture supernatant (S1B Fig) were both increased after HilD-induction. We next performed motility experiments to monitor the motility phenotype in response to HilD-activation. Surprisingly, swimming motility in soft-agar (0.3%) swim plates was drastically reduced (Fig 1B). Motility was restored to wildtype (WT) levels when a DNA binding-deficient mutant of HilD was expressed (*hilD*$_{\Delta HTH}$). Overexpression of the transcriptional regulator HilA, which is downstream of HilD in the native SPI-1 regulatory cascade, from the *araBAD* locus also resulted in decreased motility. Similarly, the deletion of *hilE*, a repressor of HilD, resulted in higher levels of SPI-1 induction (S1C Fig (left)). This was associated with a growth defect and a decrease in motility that were both restored to WT levels upon SPI-1 deletion (S1C (right) and S1D Fig). Since swimming motility in soft-agar swim plates is dependent on both bacterial growth and chemotaxis, we further assessed free-swimming motility of individual bacterial cells to exclude those effects. We observed a pronounced decrease in the single-cell swimming velocities, which decreased to a mean speed of 1.5 μm/s after 45 minutes of HilD-induction (Fig 1C (right)), which coincided with increased transcription from the HilD-dependent promoter, $P_{sicA}$. In the absence of HilD-induction (Fig 1B (left)), the mean swimming speed at that time point was 24 μm/s and the expression level of $P_{sicA}$-eGFP was only 22.5% of that in the HilD-induced cells. Moreover, after removing the inducer of *hilD* expression, swimming velocities were restored to WT levels (S1E Fig). Finally, in order to decouple production of flagella from possible HilD regulatory effects on the level of the *flhDC* promoter, we assessed motility in a strain in which the HilD binding site within the *flhDC* promoter region was randomized or deleted. Induction of HilD expression still resulted in non-motile bacteria, indicating that the observed motility phenotype was independent of a regulatory effect of HilD on *flhDC* transcription (S1F Fig).

### Adhesion factors are involved in the HilD-mediated motility defect

We next performed transcriptome analysis to assess the impact of HilD-activation on *Salmonella* gene expression on a global scale. Principal component analysis (PCA) revealed that the transcriptome of the HilD-induced strain was distinct from the control strain (S2A Fig) and we were able to identify a large number of differentially expressed genes (> 2-fold change)

under HilD-inducing conditions (S2B Fig). As expected, the expression levels of genes related to SPI-1 e.g. *prg*, *inv*, *org* and *spa* operons were increased under HilD-inducing conditions (Fig 2A). Additionally, genes of the SPI-2 operons (e.g. *sse* and *ssa*) and flagella genes (e.g., *flhD* and *fljB*) were upregulated. Interestingly, we also observed increased transcript levels of genes encoding for adhesive structures, such as SPI-4 genes, curli or other fimbrial operons (*pef* and *saf*). Moreover, the expression levels of a previously described multiprotein immunoglobulin adhesion system ZirSTU, thought to be involved in an anti-virulence pathway, were elevated (Fig 2A). The upregulation of those genes was further validated using qRT-PCR (Fig 2B). We therefore speculated that activation of HilD would induce expression of adhesive structures, which might in turn mediate the observed motility defect. We thus analysed adhesion of *Salmonella* to epithelial cells upon HilD induction in a strain deleted for SPI-1 to exclude the increased adherence that might result from the enhanced docking conferred by the activated injectisomes. Additionally, epithelial cells were pre-treated with cytochalasin D to prevent actin polymerization and bacterial invasion. Strains deleted for SPI-1 or both SPI-1 and -2 were used as control strains. As shown in Fig 2C, HilD-activation led to increased adhesion to the surface of MODE-K murine epithelial cells compared to the control strains. Accordingly, the contribution of fimbriae and curli to the HilD-dependent motility phenotype was assessed using strains overexpressing different fimbrial operons. On motility plates, only overproduction of Pef fimbriae resulted in a loss of motility, while Csg curli as well as Saf and Std fimbriae resulted in no or a very mild effect on motility (Fig 2D). However, this pronounced decrease in motility in the soft-agar swim plates might be attributed to the severe growth defect associated with the overexpression of the Pef fimbriae (S3 Fig). In contrast, the overproduction of all tested fimbriae and curli constructs resulted in a sharp decrease of single-cell swimming velocities (Fig 2E). The observed contradiction between the results obtained from the soft-agar plates and the free-swimming motility of single-cells suggests that the observed loss of motility in liquid medium is flagella-independent, but rather occurs via an adhesins-dependent manner. This can be explained by the fact that adhesins facilitate the attachment of bacteria to both cells and abiotic surfaces [53]. Thus, the glass surface of the flow chamber used in the single-cell motility experiments may provide a suitable surface for attachment of bacteria expressing adhesive structures and thereby affecting motility.

## Deletion of SPI-1 restores motility upon HilD-induction

In addition to its identified role as an activator of *flhDC* expression, HilD is known to be involved in a broad regulatory network [43]. The role of HilD as a transcriptional activator of SPI-1 gene expression remains the best characterised [21,54,55]. Other targets include SPI-2, SPI-4 and SPI-5 genes [47,49,50]. To investigate if the observed motility defect was dependent on any of those systems, we tested motility upon HilD-activation in genetic backgrounds deleted for each of them. Additionally, motility of a strain deleted for the genes encoding the twelve chaperone-usher fimbriae and curli fimbriae (Δ12 Δ*csg*) was tested. Interestingly, a ΔSPI-1 mutant displayed a WT motility behaviour upon HilD-induction on swimming motility plates (Fig 3A), as well as in liquid medium (Fig 3B). Deletions of other HilD-induced systems did not rescue the HilD-mediated motility decrease. It is noteworthy that all the strains tested in this data set expressed a sole copy of HilD under an inducible promoter, thus the feed forward loop of HilD / HilC / RtsA is disrupted, ensuring comparable levels of HilD among the strains.

## HilD induction activates the stringent response

We next investigated the growth kinetics of the bacteria upon HilD-activation using a microfluidic mother machine platform [56] (Fig 4A). Simultaneously, we determined the translational

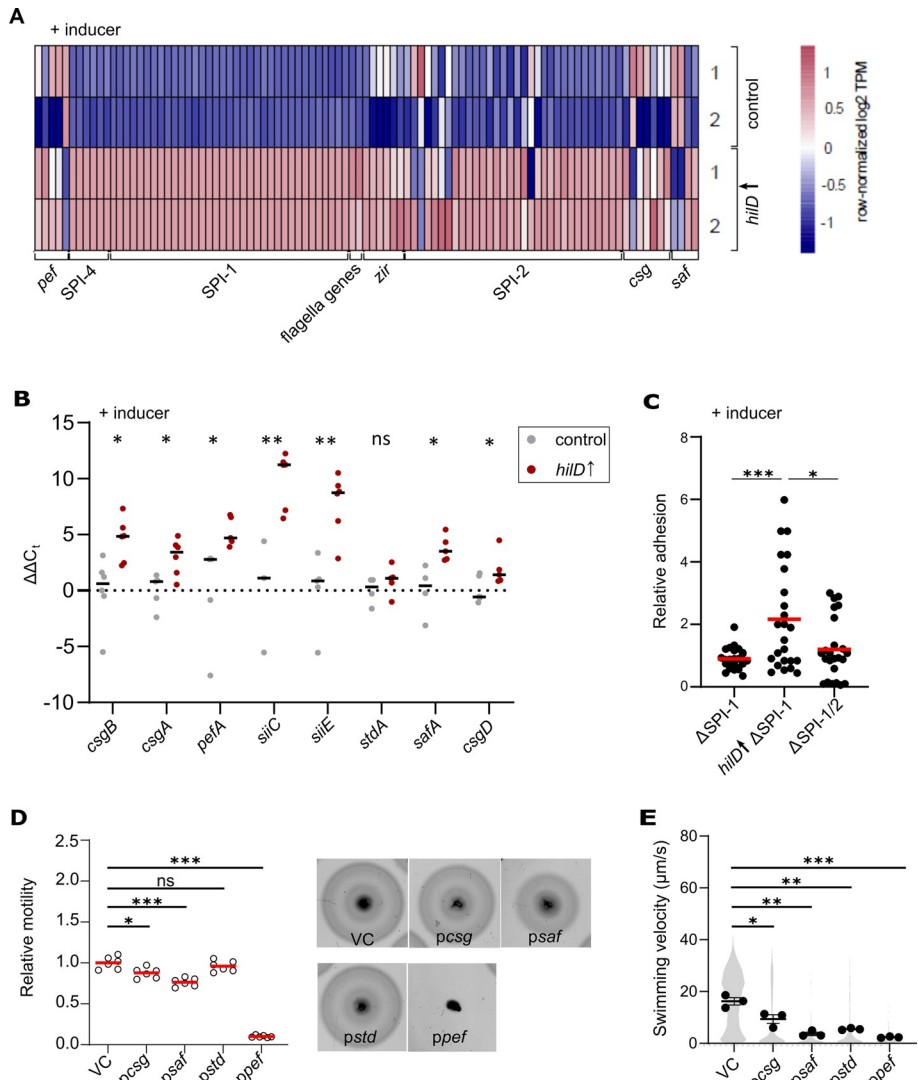

**Fig 2. HilD activates expression of adhesive structures in *Salmonella* Typhimurium.** (A) Heatmap of selected differentially expressed genes of the control (EM808) and *hilD*↑ (TH16339) strains. The TPM values of these selected genes were row normalised across the samples from two biological replicates (1, 2). (B) Validation of differential expression of *siiC*, *siiE*, *csgA*, *csgB*, *csgD*, *pefA*, *safA*, *stdA*, and *fimZ* in a HilD-induced strain compared to the control using qRT-PCR. Data from three or more biological replicates are shown as individual data points for each condition. Horizontal bars represents the calculated mean of biological replicates. Statistical significances were determined using a two-tailed Student's t-test (**, $P < 0.01$; *, $P < 0.05$;; ns, $P > 0.05$). Strains analysed were the same as in (A). (C) Relative adhesion of different *Salmonella* strains to MODE-K murine epithelial cells. MODE-K cells were incubated with different ΔSPI-1 strains at a MOI of 10 for 1 h at 37°C. After extensive washing, cells were lysed and plated for CFU assessment. Counted CFU values were normalized to the ΔSPI-1 strain as a control. Data from 24 biological replicates are shown as individual data points. Horizontal bars (red) represent the calculated mean of biological replicates. Statistical significances were determined using a two-tailed Student's t-test (***, $P < 0.001$; *, $P < 0.05$). Strains analysed were EM830 (ΔSPI-1), EM93 (*hilD*↑ ΔSPI-1) and EM829 (ΔSPI-1/2). *hilD*↑: strain expressing *hilD* under an inducible promoter. (D) Swimming motility of strains overexpressing different adhesins in *trans* in soft-agar swim plates was monitored at 37°C for 3.5 h. Diameters of swimming halos were measured and normalized to the control strain (left). Representative swimming halos of the analysed mutants are shown (right). Data from six biological replicates are shown as individual data points. Horizontal bars (red) represent the calculated mean of biological replicates. Strains analysed were EM12144 (VC, vector control), EM12145 (p*csg*), EM12146 (p*saf*), EM12147 (p*std*), EM12148 (p*pef*). Statistical significances were determined using a two-tailed Student's t-test (***, $P < 0.001$; *, $P < 0.05$; ns, $P > 0.05$). (E) Single-cell swimming velocities of strains overexpressing different adhesins. Individual data points represent the averages of the single-cell velocities of independent experiments. Violin plots represent data values from at least 700 analysed single-cell tracks. Horizontal bars (bold) represent the mean of the calculated average velocities of three independent experiments. The error bars represent the standard error of mean and statistical significances were determined using a two-tailed Student's t-test (***, $P < 0.001$; **, $P < 0.01$; *, $P < 0.05$). Strains analysed were the same as in panel (D). AnTc: anhydrotetracycline.

capacity of the cells by monitoring the fluorescence intensities resulting from the translation of a GFP expressed under an arabinose inducible promoter. Induction of HilD was associated with a growth defect as reported before [57], as well as a decreased translation rate (Figs 4A and S4A). Phase contrast microscopy revealed a reduction of the average cell length upon HilD-activation to ~1.5 μm compared to ~2.5 μm in WT cells, as well as a change in morphology from rod-shaped to coccoid (S4B Fig). Results from the mother machine platform confirmed that HilD-activation led to a ~49% decrease in cell elongation rate (Figs 4B and S4C (left)). The average maximal cell length during a division cycle decreased from 3.4 μm in the control cells to 2.2 μm in the HilD-induced cells. Consistently, the increase of cell length during a division cycle was ~1.1 μm in HilD-induced cells compared to ~1.5 μm in the control cells (Figs 4B and S4C (middle and right)). The analysis further demonstrated an increase of the average generation time to ~54 min upon HilD-induction compared to ~28 min in the control (Fig 4C (left)). Although the mean GFP fluorescence intensity in the HilD-induced cells dropped to almost 47% of that of the control, the cells retained their translational capacity through the duration of the experiment (Fig 4A and 4C (right)). Interestingly, deletion of SPI-1 was able to restore the growth rate defect, as well as cell length and morphology to the WT (S4A and S4B Fig). We therefore speculated that HilD-induction results in a cellular state that mimics a nutrient-limited environment. To investigate this possibility, we constructed a translational fusion of the fluorescent protein mCherry to *rpoS* in order to monitor activation of the stringent response. Expression of *rpoS* was enhanced in cells overproducing a hyperactive variant of the (p)ppGpp synthase RelA [58,59], validating the functionality of the stringent response reporter (S5A Fig). Additionally, *rpoS* levels were elevated when the cells were incubated in minimal medium devoid of carbon and nitrogen sources compared to their levels in the nutrient-rich LB medium (S5B Fig). We therefore investigated the levels of *rpoS* expression as a proxy for the stringent response in HilD-induced cells. As shown in Fig 4D, *rpoS* expression levels were elevated in HilD-induced cells compared to the control strain and decreased back to WT levels upon SPI-1 deletion, suggesting that HilD-activation induces the stringent response.

## HilD induction is not associated with loss of flagellation

The anti-FlhDC factor, RflP (formerly known as YdiV) was shown before to tune the expression of flagella genes in response to nutrient levels [42]. Hence, we hypothesized that a nutrient-limited status after HilD-activation in addition to the general defect in translation result in enhanced *rflP* expression, which would act on FlhDC at the post-translational level catalysing its proteolytic degradation, thereby repressing flagella synthesis and motility. Subsequently, we investigated the levels of RflP expression using a translational fluorescent protein reporter fusion to *rflP*. As shown in S6 Fig, we observed a small, yet statistically non-significant, increase in RflP expression upon HilD-activation, that was restored to WT levels in cells deleted for SPI-1. This result indicated that flagella synthesis might be impaired in response to the HilD-induced nutrient-limited cellular state. We therefore performed flagella immunostaining in order to evaluate the flagellation state upon HilD-activation. However, as shown in Fig 5, quantification of flagella numbers per cell demonstrated that HilD-induced cells remained flagellated, albeit the average number of flagella per cell in HilD-induced cells was slightly decreased compared to the control strain. Additionally, we tested the expression levels of different flagellar proteins in presence and absence of HilD-induction by Western blotting. The tested proteins included the basal body-associated protein FliG, as well as the molecular ruler FliK and the flagellin FliC, which are secreted at different stages of flagella assembly via the flagellar T3SS, respectively. The results of the Western blot analysis confirmed that HilD-induced cells are still expressing and secreting flagellar proteins (S7 Fig).

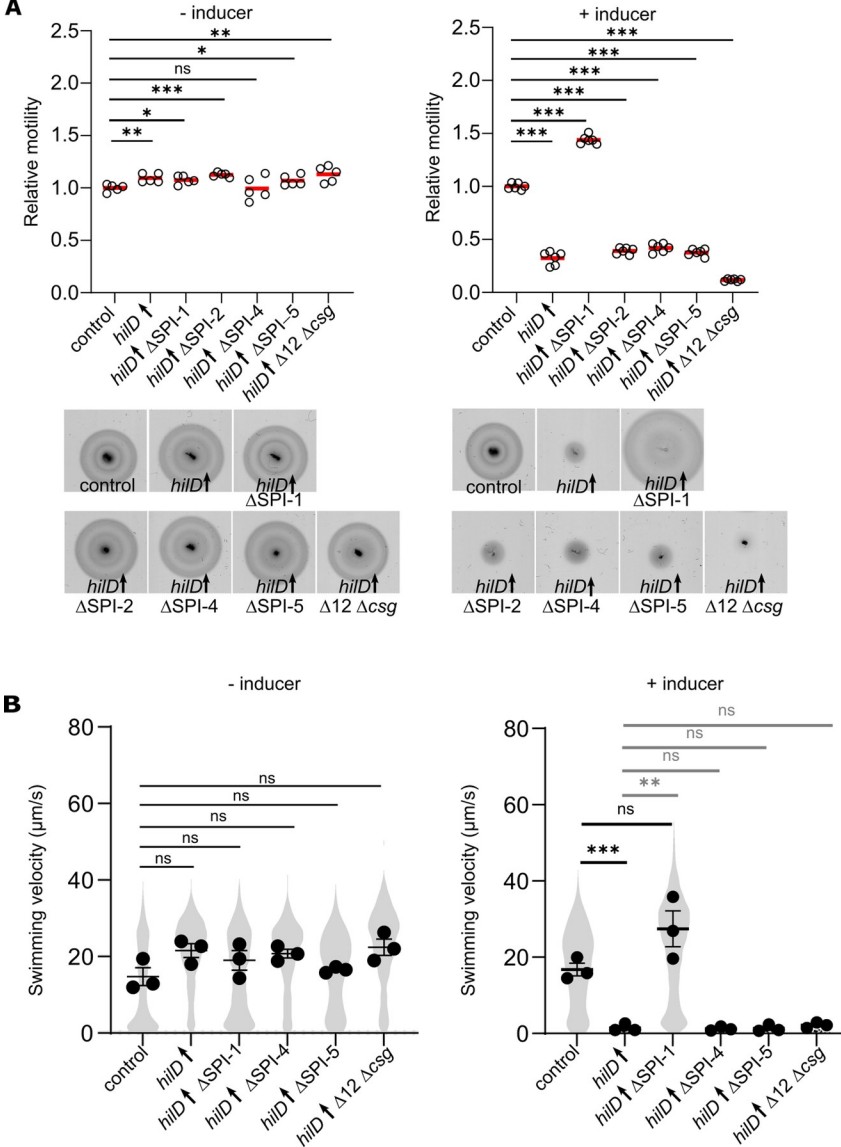

**Fig 3. Deletion of SPI-1 restores motility upon HilD induction.** (A) Swimming motility of the control, *hilD*↑ and various SPIs/adhesins mutant strains in soft-agar swim plates was monitored at 37˚C for 3.5 h in absence (upper panel) or presence (lower panel) of 100 ng/ml AnTc to induce HilD production. Diameters of swimming halos were measured and normalized to the control strain (left). Representative swimming halos of the analysed mutants in absence (upper panel) or presence (lower panel) of AnTc are shown (right). Data from five biological replicates are shown as individual data points. Horizontal bars (red) represent the calculated mean of biological replicates. Statistical significances were determined using a two-tailed Student's t-test (***, $P < 0.001$; **, $P < 0.01$; *, $P < 0.05$; ns, $P > 0.05$). (B) Single-cell swimming velocities of the control, *hilD*↑ and various SPIs/adhesins mutant strains in absence or presence of AnTc to induce HilD production. Individual data points represent the averages of the single-cell velocities of independent experiments. Violin plots represent data values from at least 700 analysed single-cell tracks. Horizontal bars (bold) represent the mean of the calculated average velocities of three independent experiments. The error bars represent the standard error of mean and statistical significances were determined using a two-tailed Student's t-test (***, $P < 0.001$; **, $P < 0.01$; ns, $P > 0.05$). Strains analysed were TH437 (control), TH17114 (*hilD*↑), EM12479 (*hilD*↑ ΔSPI-1), EM13020 (*hilD*↑ ΔSPI-2), EM12354 (*hilD*↑ ΔSPI-4), EM13021 (*hilD*↑ ΔSPI-5) and EM12648 (*hilD*↑ Δ12 Δ*csgA*). *hilD*↑: strain expressing *hilD* under an inducible promoter. Δ12: strain deleted for 12 chaperone-usher fimbrial operons. AnTc: anhydrotetracycline.

### Dissipation of the membrane potential contributes to the HilD-mediated motility defect

The proton motive force (PMF) serves as a main source of energy that drives secretion of protein substrates via the T3SS of both the flagellum and the injectisome [7,60]. Additionally, it energizes the rotation of the flagellar motor, thereby enabling swimming motility [61,62]. Therefore, we reasoned that increased secretion of injectisome substrates might dissipate the available PMF pool, thereby affecting flagella rotation and motility. Further, we recently reported that short-term starvation dissipates the PMF [63]. Similarly, it was previously described by Verstraeten et al. that nutrient starvation induces a stringent response resulting in PMF dissipation in *Escherichia coli* [64]. Altogether, this suggests that induction of the stringent response observed upon HilD-activation might affect the membrane potential. We thus examined the membrane potential status in HilD-induced cells using the membrane potential-sensitive dye DiSC$_3$(5) [63]. As a control, cells were treated with the ionophore CCCP prior to staining in order to dissipate the PMF. As expected, CCCP treated cells displayed diminished DiSC$_3$(5) signal intensity (S8 Fig). Interestingly, HilD-activation resulted in a sharp decrease of the membrane potential, which was largely restored upon SPI-1 deletion (Fig 6). These results indicate that dissipation of PMF caused by increased assembly and/or activity of the injectisome and other PMF-draining systems might contribute to the observed motility defect.

## Discussion

Flagella-mediated motility is crucial for *Salmonella* to move efficiently through the host intestinal lumen and establish a successful infection. Production of flagella and the associated chemotaxis system requires expression of more than 60 genes [65] that are tightly regulated in response to the different environmental cues. Additionally, the flagellum is interconnected to the virulence-associated injectisome via a complex cross-regulatory network (Fig 1A). A previously characterized crosstalk involves the transcriptional activation of the flagellar master regulatory operon *flhDC* by HilD, the master regulator of SPI-1. Here, we characterized the motility phenotype associated with the induction of HilD.

Unexpectedly, we observed a pronounced motility defect after HilD induction, while the cells surprisingly remained flagellated (Figs 1 and 5). We showed that motility was restored only in mutants lacking the SPI-1 locus, but not other SPIs. A transcriptome analysis upon HilD-induction revealed several upregulated genes that might contribute to the observed motility phenotype. As expected, genes related to SPI-1, SPI-2, SPI-4 and flagella were upregulated upon HilD induction. The genes belonging to SPI-1, SPI-2 and SPI-4 were similarly reported to be positively regulated by HilD by Smith et al. and Colgan et al., while only Smith et al. reported the upregulation of *flhDC* upon HilD overexpression [66,67]. The gene encoding the methyl-accepting chemotaxis proteins McpA as well at the chemotaxis genes CheB/Y/Z were also upregulated as described previously [67,68]. Moreover, various uncharacterized genes (STM05010, STM1329, STM1330, STM1600, STM1854, STM2585, STM4079s, STM4310, STM4312 and STM4313) were positively regulated by HilD, as described before by Smith et al. and Colgan et al. [66,67]. Interestingly, our transcriptome analysis revealed additional differentially regulated genes that were not described in previous studies [43,66], likely because of the prolonged time of HilD induction employed in the current study. Colgan et al. compared the transcriptomic profile of a *hilD* deletion strain to the WT, while Smith et al. used pulse expression of HilD from a plasmid in a *hilD* deletion mutant before analyzing the transcriptome compared to a deletion strain carrying an empty vector. While these approaches enabled the identification of genes directly regulated by HilD, the longer HilD induction time used here enabled us to also identify genes that were indirectly affected by HilD presence.

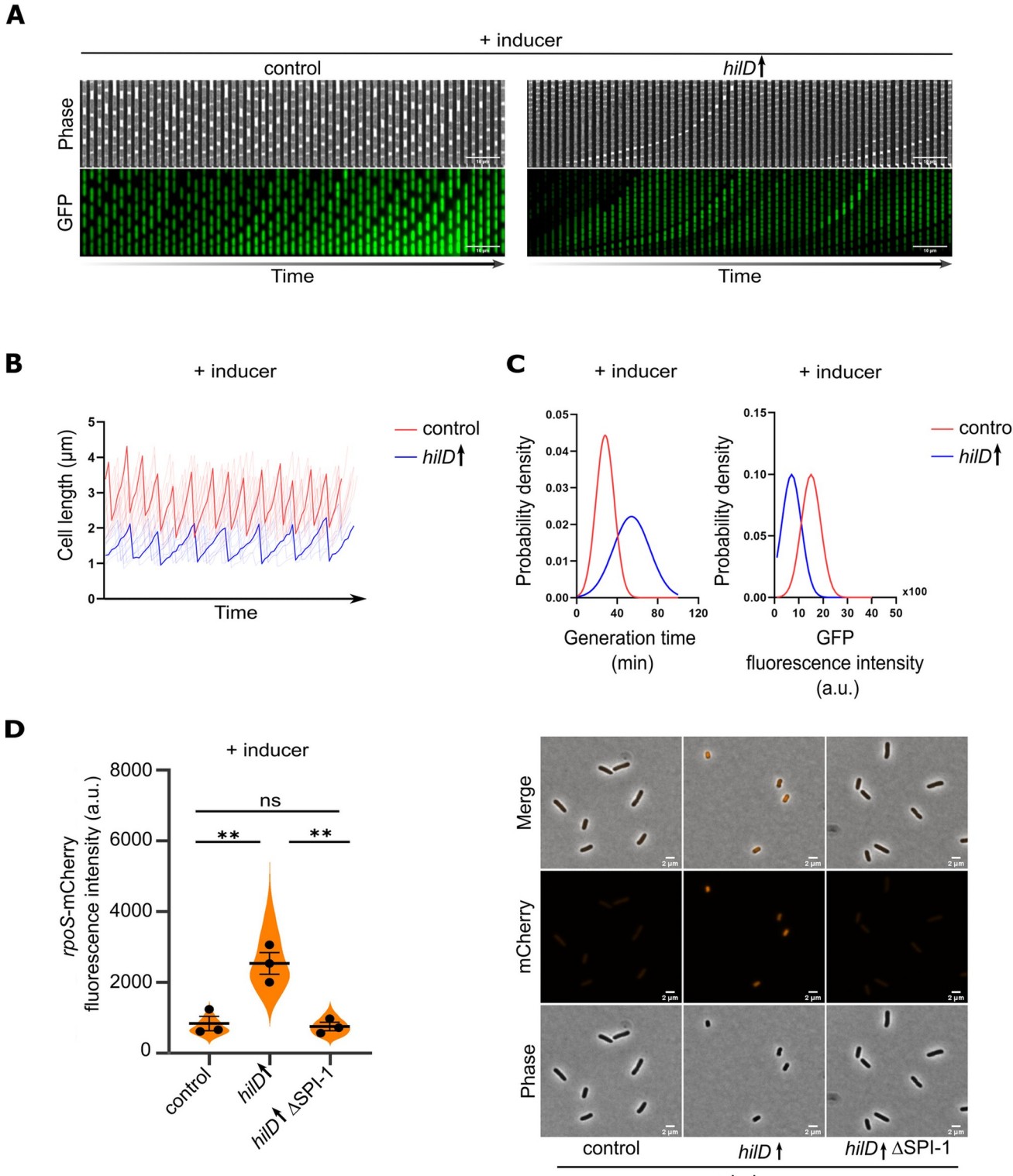

**Fig 4. HilD induction activates the stringent response.** (A) HilD-activation results in decreased growth rate and decreased translation rate. Strains were grown in the presence of AnTc as an inducer for HilD production. The expression of pBAD-GFP was achieved by addition of arabinose. The kymographs illustrate a single lineage of HilD-induced or control cells. Images from every second frame from the time-lapse were used to create the shown sequence. (B) Changes in cell length through time as determined using time-lapse microscopy for individual lineages. Traces of a sample of ten lineages are plotted in translucent colour and a sample trace is overlaid in opaque colour for each strain. (C) Probability density function of generation times (left) and GFP

translation levels (right) in single cells as determined by time-lapse microscopy. Strains analysed in (A), (B) and (C) were EM12802 (control) and EM12803 (*hilD*↑). Violin plots represent at least 300 single-cell data values from a representative experiment. *hilD*↑: strain expressing *hilD* under an inducible promoter. a.u.: arbitrary units. (D) HilD induction activates the expression of *rpoS*; a reporter for the stringent response. Fluorescence intensities of a *rpoS*-mCherry translational fusion were quantified using fluorescence microscopy (left). Individual data points represent the averages of the single-cell values of independent experiments. Violin plots represent data values from at least 700 analysed single cells. Horizontal bars (bold) represent the mean of the calculated average of three independent experiments. The error bars represent the standard error of the mean and statistical significances were determined using a two-tailed Student's t-test (**, $P < 0.01$; ns, $P > 0.05$). Representative microscopy images are shown (right). Scale bar is 2 μm. Strains analysed were EM13017 (control), EM13065 (*hilD*↑) and EM13276 (*hilD*↑ ΔSPI-1). *hilD*↑: strain expressing *hilD* under an inducible promoter. a.u.: arbitrary units.

Accordingly, we could detect a number of upregulated genes that encode adhesive structures. (Fig 2). These adhesin-related genes include curli fimbriae genes encoded by the *csg* operons, the chaperone-usher fimbriae Pef and Saf as well as a multiprotein immunoglobulin adhesin system encoded by the *zirTSU* operon [69–80]. Consistent with the upregulations of adhesin operons, we observed an increased adhesion of *Salmonella* to intestinal epithelial cells upon HilD-activation (Fig 2C), which might explain the decreased swimming velocities of single cells (Figs 1B and 2E). Further, overproduction of adhesin systems phenocopied the swimming defect of the HilD-induced strain. It is thus tempting to speculate that *Salmonella* induces both the SPI-1 encoded injectisome and several adhesin systems during the initial stage of infection in order to enhance binding of the bacteria to epithelial cells.

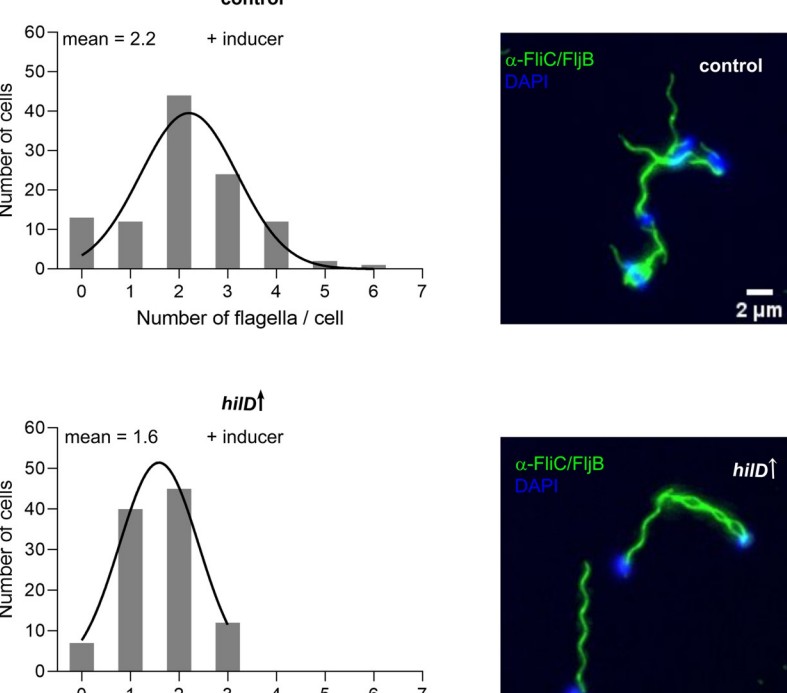

**Fig 5. HilD induction does not affect flagellation.** Counts of flagella per cell were determined using fluorescence microscopy. Histograms of counted flagella per cell are shown to the left. Average flagella numbers were calculated by Gaussian non-linear regression analysis (black line). The plotted data represent at least 100 single cells from one representative experiment. Representative microscopy images of flagella immunostaining are shown to the right. Scale bar is 2 μm. Strains analysed were TH437 (control) and TH17114 (*hilD*↑). *hilD*↑: strain expressing *hilD* under an inducible promoter.

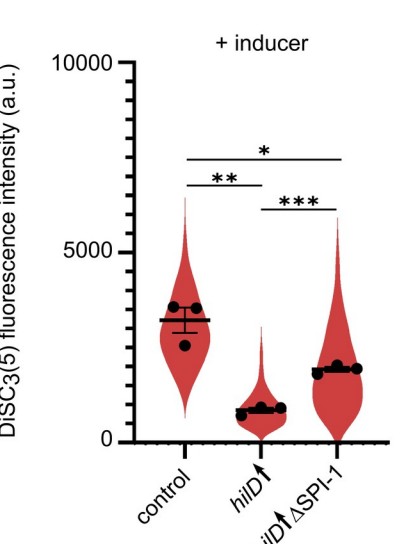
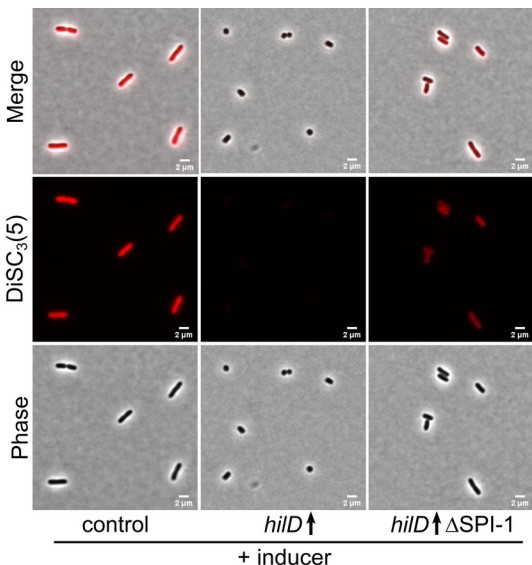

**Fig 6. HilD induction results in membrane depolarization.** Strains were grown under HilD-inducing conditions using AnTc followed by staining with the membrane potential sensitive dye $DiSC_3(5)$. $DiSC_3(5)$ fluorescence intensities of single cells were quantified (left). Individual data points represent the averages of the single-cell data of independent experiments. Violin plots represent data values from at least 400 analysed single cells. Horizontal bars (bold) represent the mean of the calculated average of three independent experiments. The error bars represent the standard error of mean and statistical significances were determined using a two-tailed Student's t-test. The error bars represent the standard error of the mean and statistical significances were determined using a two-tailed Student's t-test (***, $P < 0.001$; **, $P < 0.01$; *, $P < 0.05$). Representative microscopy images are shown (right). Scale bar is 2 μm. Strains analysed were TH437 (control), TH17114 (*hilD*↑), EM12479 (*hilD*↑ ΔSPI-1), *hilD*↑: strain expressing *hilD* under an inducible promoter. AnTc: anhydrotetracycline. a.u.: arbitrary units.

In line with previous studies, we additionally observed a growth defect upon HilD induction [57]. This growth defect was evidenced by a decrease in the cell elongation rate, the maximal observed cell length and an increased generation time. The growth retardation was associated on a cellular level with morphological changes from rod-shaped to coccoid. The deletion of SPI-1 was able to restore the growth rate and morphology of the cells. Similarly, Hockenberry et al. reported a 25% decrease in growth rate in a population of cells expressing SPI-1 genes compared to a non-expressing population [81]. This decrease in growth rate was also associated with decreased cell length. Arnoldini et al. reported a drop of the doubling rate from 0.96 to 0.26 in SPI-1 expressing cells [82]. Although the quantification results vary among the different studies including the current one, they all confirm a growth penalty associated with SPI-1 gene expression. The differences are likely due to the different experimental setups used. In the current study, we employed a continuous culture setup where fresh medium is continually pumped into the chip. Hockenberry et al., investigated the growth rate of single cells by using medium from a simultaneously growing batch culture. Arnoldini et al. used spent medium, which exposes the cells to medium with depleted nutrients and a variety of signalling molecules from the start of the experiment. The morphological changes upon HilD-activation together with the observed decrease in the cell translational capacity suggested that the cells were undergoing a state of starvation [83–85]. This was further evidenced by the upregulation of *rpoS* expression (Fig 4), which has previously been used as a proxy for induction of the stringent response [86,87]. Although nutrient deprivation is known to activate the anti-FlhDC factor RflP [88], we found that the HilD-induced cells remained flagellated (Fig 5).

Interestingly, a similar phenotype has previously been observed in *Rhizobium melitoli*, where starved cells lost motility while retaining normal flagellation [89]. Additionally, previous studies suggested that starvation conditions and induction of the stringent response result in dissipation of the membrane potential [63,64,90], which is critical for energizing flagellar rotation and motility. It has also been reported that mis-localisation of the multimeric *invG*-encoded secretin ring of the injectisome into the inner membrane, instead of the outer membrane, results in membrane perturbations [91,92]. This in turn induces the phage shock protein (Psp) system, which functions to maintain the PMF and the barrier function of the inner membrane. Recently, Sobota et al. observed a decrease in the PMF of cells expressing SPI-1 after exposure to external stress agents, while maintaining their PMF levels in its absence [93]. In the current study, we observed a decrease in membrane potential upon HilD-induction, which was restored by SPI-1 deletion (Fig 6).

The physiological consequences of the cross-regulation between the flagellum and the virulence-specific T3SSs (vT3SS) of the different organisms remains elusive. For example in the enterohemorrhagic *Escherichia coli* (EHEC) which is closely related to *Salmonella*, the constitutive expression of *flhDC* results in decreased adherence to epithelial cells. In line with that, a reciprocal cross-regulation between the flagellum and the vT3SS was described, where the transcriptional regulator GrlA represses the transcription from different flagellar promoter classes while positively regulating the vT3SS encoding genes [94]. Similarly, the GacA response regulator activates the flagellar gene expression but represses the expression of the vT3SS genes in *Pseudomonas aeruginosa* [95]. Bleves et al. also reported the upregulation of the virulon transcriptional regulator VirF in the absence of *flhDC* in the more distantly related *Yersinia enterocolitica*, in addition to the upregulation of *ysc* and *yop* genes encoding the structural components and effector proteins of the vT3SS, respectively [96]. However, although some reciprocal cross-regulation is also observed in *Salmonella*, flagella seem to play an important role not only in the motility process but also in the adherence and invasion to epithelial cells in contrast to what Iyoda et al. reported in EHEC [4,5,40]. This might explain why HilD-induced cells are still expressing flagellar genes even though they are non-motile.

In summary, we speculate that it might be important for *Salmonella* to quickly modulate its swimming motility upon reaching its target sites on the host cells in order to facilitate docking to the cells and to enable subsequent injection of effector proteins. Upon reaching the small intestine, *Salmonella* undergoes a phase of chemotaxis and flagellar motility inside the intestinal lumen in search for the intestinal epithelial cells [63]. Afterwards, the bacteria perform a process known as near-surface swimming, where they screen the epithelial cell surface for permissive entry sites to start the invasion process [97,98]. The environmental cues in the small intestine triggers the expression of HilD and the subsequent assembly of the SPI-1 encoded injectisomes and effector proteins, in addition to the other adhesive structures [21,48,99]. The secretion of effectors, however, remains minimal until *Salmonella* reaches its target site on the epithelial cell surface, upon which effectors are actively secreted via the SPI-1 encoded injectisome to mediate membrane ruffling [100–102]. This, consequently, results in more injectisomes becoming actively engaged in the secretion process. It has also been suggested that upon contact with the host cell, additional injectisomes are assembled to meet the sudden increase in effector secretion rate [103]. Considering the crucial role of flagella in *Salmonella* pathogenesis, not only for motility but also for facilitating adhesion to and invasion into host epithelial cells [4,5], a mechanism to eject already produced flagella, as recently described for the polarly flagellated γ-proteobacteria [104], might be disadvantageous. It is plausible that *Salmonella* would prefer a strategy that maintains flagellation even during stages of the infection process where active motility is less crucial. For instance, the presence of flagella might enhance bacterial adherence to host cells and facilitate the invasion process by inducing actin

polymerization [4,5]. An additional advantage of retaining but downregulating flagella function upon reaching the target entry site could be the ability of *Salmonella* to quickly restart the infection process in case of an unsuccessful host cell infection. In conclusion, we propose a model in which *Salmonella*, upon HilD-mediated SPI-1 induction, employs a complex process to upregulate adhesive structures and dissipate the membrane potential. This rapidly halts motility and prepares the bacteria for efficient host cell invasion upon reaching the target entry sites on the host epithelial cells (Fig 7).

## Materials and methods

### Strains, media and bacterial growth

All *Salmonella enterica* serovar Typhimurium strains are listed in S1 Table. Bacteria were routinely grown in lysogeny broth (LB) at 37°C, supplemented with 100 μg/ml ampicillin, 10 μg/ml chloramphenicol, 25 μg/ml kanamycin, 100 ng/ml anhydrotetracycline (AnTc) or 0.2% arabinose when needed. Mutant strains were constructed using the general transducing *Salmonella* phage P22 HT105/1 int-201 [105] or using λ-RED recombination [106,107]. Bacterial growth was assessed via measurement of optical density at 600 nm ($OD_{600}$) in a microplate reader (Tecan).

### Plasmid construction

To construct the plasmid expressing truncated RelA, the region coding for the N-terminal 455 amino acids of *relA* was PCR amplified from the genomic DNA of a wild type *Salmonella* strain (TH437) using cgccatatgGTCGCGGTAAGAAGTGCACA as a forward primer and ctagtctagatcattattaCAACTGATAGGTGAATGGCA as a reverse primer. The primers were designed with overhangs (lowercase letters) containing restriction sites (underlined) for *Nde*I and *Xba*I enzymes, respectively. The digested product was cloned into the *Nde*I and *Xba*I sites on the pTrc99a-FF4 plasmid under the control of the IPTG-inducible *trc* promoter.

### Swimming motility

Swimming motility was assessed by inoculating 2 μl of overnight cultures of the desired strains into soft-agar swim plates containing 0.3% agar followed by incubation at 37°C for approximately 3.5 h. For testing swimming motility under SPI-1 inducing conditions (S1D Fig), the NaCl concentration in the plates was adjusted to 1%; a condition previously shown to induce SPI-1 gene expression [21,108].

### Single cell tracking

Bacteria were grown until mid-exponential growth phase in the presence of supplements where needed, then diluted into fresh LB to an OD600 ≤ 0.1. An aliquot was subsequently transferred to a flow chamber and a time-lapse video was recorded using a Nikon Eclipse Ti2 inverted microscope equipped with a CFI Plan Apochromat DM 20× Ph2/0.75 objective (Nikon) at 70 msec intervals. The obtained videos were segmented using the pixel classifier in Ilastik v1.3.3 [109] to recognize background and cells. Subsequently, all images in the dataset were processed to produce a "Simple Segmentation" mask output and exported to Fiji [110], where trajectories of the bacteria were tracked and the velocities of single cells were calculated using the simple LAP tracker of the Fiji plugin TrackMate [111].

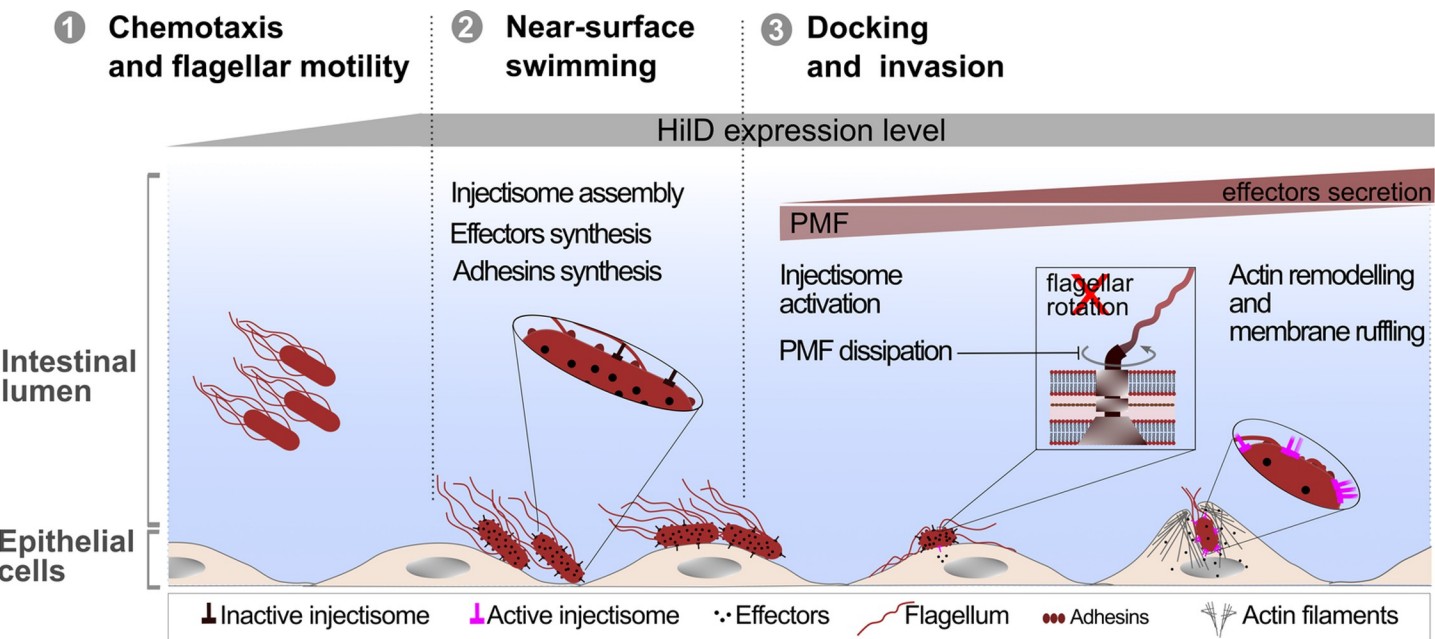

**Fig 7. HilD-mediated SPI-1 induction abrogates motility.** Inside the intestinal lumen, *Salmonella* goes through a phase of chemotaxis and flagellar motility in search for the host epithelial cells. The intestinal environment also triggers HilD expression (1). Upon reaching the epithelial cells, *Salmonella* undergoes a phase of near-surface swimming in search for the permissive entry sites on the cells. HilD-induced cells are now equipped with assembled SPI-1 encoded inectisomes, effector proteins and adhesins (2). After landing on the target site, the injectisomes are activated and start effectors secretion (3). SPI-1 induction upregulates adhesive structures and dissipates the membrane potential, therefore rapidly abrogating motility, enhancing docking to host cells and priming the bacteria for efficient host cell invasion.

## Fluorescence microscopy

For live cell microscopy, cultures were spotted on the surface of 1% agarose pads (in PBS) cast on SuperFrost Plus slides (Erpedia). The spotted cultures were allowed to air-dry briefly, then covered with a microscopy cover slip (1.5H, Roth). For fixed cells imaging, in-house flow-chambers were constructed using slides and cover slips treated with 0.1% poly-L-lysine (Sigma-Aldrich) [112]. The slide and cover slip were assembled in the presence of a double layered parafilm as a spacer. Samples were loaded into the chamber and allowed to adhere to the cover slip at room temperature (RT) followed by fixation with 4% paraformaldehyde for 10 min. The fixed cells were then washed with PBS and mounted in Fluoroshield mounting medium containing DAPI (Sigma-Aldrich). Image acquisition was carried out using a Nikon Eclipse Ti2 inverted microscope equipped with a CFI Plan Apochromat DM 60× Lambda oil Ph3/1.40 objective (Nikon). The filter cube LED-CFP/YFP/mCherry-A (CFP / YFP / mCherry —Full Multiband Triple) (Semrock) was used for imaging *rpoS*-mCherry fusions and LED-DA/FI/TR/Cy5/Cy7-A (DAPI / FITC / TRITC / Cy5 / Cy7—Full Multiband Penta) (Semrock) was used for P$_{sicA}$-eGFP, *rflP*-mScarlet, DAPI and DiSC$_3$(5).

## Microfluidic chip fabrication

Custom microfluidic master molds fabricated with Electron Beam Lithography were ordered from ConScience AB (Sweden). The mother machine chip used in this study features micro-channels of 30 μm length, 1 μm width and 0.8 μm height. To prepare the PDMS chips, SYL-GARD 184 silicone elastomer base (Dow, Europe) was mixed with the curing agent at a 7:1 ratio, degassed under vacuum for 30 minutes and poured onto the wafer and cured overnight at 80°C. Inlets and outlets were punched in the chip using a disposable biopsy punch of 0.75

mm diameter. The PDMS chip was bonded to a high precision cover glass (24 x 60mm) using a 30 seconds oxygen plasma treatment in a plasma cleaner (Deiner) and incubated at 80°C for 10 minutes.

## Microfluidic experimental setup and data analysis

Bacteria were grown in LB supplemented with ampicillin until late exponential phase. Cells were then loaded into the microchannels with a syringe tip connected to a syringe with Tygon tubing (inside diameter 0.51 mm and outside diameter 1.5 mm). LB supplemented with ampicillin, arabinose and AnTc was used as a growth medium and was delivered into the chip via Tygon tubbing connected to the inlets, using a syringe pump (World Precision Instruments). First the flow rate was set to 45 μl/min for 15 min to allow the main channels to clear, and then reduced to 5 μl/min for the duration of the experiment. The flow-throw was collected from the outlet with a second Tygon tubbing (inside diameter 0.51 mm and outside diameter 1.5 mm). Images were taken every 5 minutes for 20 hours. The automatic focalization was obtained via the Nikon PerfectFocus system. Bacmann software implemented in Fiji, was used for bacterial cell segmentation and tracking [81]. After initial pre-processing of the data (segmentation and rotation), centers of the microchannels were detected using the GFP channel, further allowing channel segmentation "*MicrochannelTracker* with *MicroChannelFluo2D* Segmenter". Furthermore, the GFP channel was used to subsequently detect bacterial cells "*BacteriaClosedMicrochannelTrackerLocalCorrections* with *BacteriaFluo* module". Time frames before the fluorescence signal reached a level sufficient for cell segmentation were excluded from further analysis. The resulting data were further analysed in Python 3.0.

## Flagella staining

Logarithmically grown cells were fixed on poly-L-lysine pre-coated coverslips as described previously followed by incubation with a solution of 10% BSA to block non-specific antibody binding. The flagellar filament was immunostained by incubating with a 1:1 mixture of α-FliC and α-FljB primary antibodies (Difco) diluted 1:1,000 (in 2% BSA solution). Samples were subsequently incubated with 10% BSA to block the unspecific binding followed by incubation with a secondary α-rabbit antibody conjugated to Alexa Fluor 488 (Invitrogen) and diluted 1:1,000 (in PBS). Samples were mounted in Fluoroshield supplemented with the nucleic acid stain DAPI (Sigma-Aldrich).

## DiSC$_3$(5) membrane potential assay

Logarithmically grown cultures were diluted to $OD_{600}$ of 0.15 in LB medium. A volume of 500 μl was transferred to a 2 ml round bottom Eppendorf tubes, BSA to 0.5 mg/ml was added followed by the addition of the membrane potential sensitive dye 3,3'-Dipropylthiadicarbocyanine iodide (DiSC$_3$(5)) [63]. The suspensions were allowed to incubate under shaking conditions in a thermomixer for 5 min. The lids of the Eppendorf tubes were left open to maintain sufficient aeration. Aliquots of 1 μl were then transferred to 1% agarose pads (in PBS) and imaged immediately using Nikon Eclipse Ti2 inverted microscope where samples were excited at 647 nm for the detection of DiSC$_3$(5) fluorescence. For validation of the DiSC$_3$(5) dye, the diluted cultures were treated with the protonophore carbonyl cyanide m-chlorophenyl hydrazone (CCCP) at a final concentration of 1 mM for 15 min before adding the dye. Segmentation of bacterial cells and quantification of DiSC3(5) fluorescence intensities were performed using the Fiji plugin MicrobeJ [113].

## Microscopic evaluation of *rpoS*-mCherry reporter fusion

To validate the constructed *rpoS*-mCherry fusion in response to starvation conditions, a WT strain expressing the fusion was grown until mid-exponential phase under shaking conditions at 37°C in LB medium. Subsequently, the cells were centrifuged and resuspended in either M9 medium (Difco) supplemented with 1.0 mM $CaCl_2$ and 0.1 mM $MgSO_4$ but devoid of carbon or nitrogen sources or LB. The cells were then incubated for one additional hour. Aliquots were transferred to 1% agarose pads for imaging. For validating the fusion in response to elevated levels of (p)ppGpp, a *Salmonella* strain carrying a plasmid expressing a truncated RelA comprised of the 455 N-terminal amino acids under an IPTG-inducible promoter was grown in LB supplemented with ampicillin under shaking conditions at 37°C for approximately 2 h. Afterwards, IPTG was added to the culture to a final concentration of 0.2 mM and incubated for an additional hour. Subsequently, aliquots were transferred to 1% agarose pads for imaging.

## Kinetic measurements of HilD-induced changes in single-cell swimming velocities and $P_{sicA}$-eGFP expression

To monitor the effects of HilD induction on the swimming velocities and the expressed levels of the SPI-1 promoter fusion ($P_{sicA}$-eGFP) in single bacterial cells, strains were grown at 37°C to exponential phase in LB. Subsequently, 100 ng/ml anhydrotetracycline (AnTc) was added to induce HilD. Samples were taken immediately, and incubation of the cultures was resumed. Additional samples were then taken every 15 min. Single-cell swimming velocities were measured as described above and $P_{sicA}$-eGFP fluorescence intensities were measured in the cultures using a microplate reader (Tecan). The measured fluorescence intensities were normalized to the $OD_{600}$ of the cultures and reported as relative fluorescence units (RFU).

## Measurements of growth and $P_{sicA}$-eGFP expression under physiological SPI-1 inducing conditions

To monitor the growth kinetics and $P_{sicA}$-eGFP expression under physiological SPI-1 conditions, bacteria were grown in 1% NaCl containing LB. Growth was assessed via measurement of optical density at 600 nm ($OD_{600}$) in a microplate reader (Tecan). $P_{sicA}$-eGFP fluorescence intensities were also measured in the cultures in the microplate reader. The measured fluorescence intensities were normalized to the $OD_{600}$ of the cultures and reported as relative fluorescence units (RFU).

## Microscopic evaluation of HilD-activation

In order to validate the activity of HilD under control AnTc or arabinose inducible promoters, the levels of the HilD-dependent $P_{sicA}$-eGPF transcriptional fusion were assessed as a reporter for SPI-1 genes activation. Strains expressing the $P_{sicA}$-eGPF fusion and *hilD* under an inducible promoter were grown exponentially in absence or presence of 100 ng/ml AnTc or 0.2% arabinose, respectively. Samples were transferred 1% agarose pads and imaged as described above. Segmentation of bacterial cells and quantification of $P_{sicA}$-eGPF fluorescence intensities were performed using the Fiji plugin, MicrobeJ [113].

## Adhesion assay

The murine epithelial cell line MODE-K was used for adhesion assays. $2.5 \times 10^5$ cells/ml were seeded in 24-well plates and pre-treated with 1 μg/ml cytochalasin D for 30 min to inhibit actin polymerization. Cells were incubated with *Salmonella* strains at an MOI of 10 for 1 h. Afterwards the epithelial cells were washed extensively with PBS to remove non-adherent

bacteria. Epithelial cells were subsequently lysed with 1% Triton X-100 and cell lysates were serially diluted and plated on LB plates to determine the CFU/ml as a measure of bacteria that adhered to the epithelial cells. All values were normalized to the control strain.

### Secretion assay

*Salmonella* cultures were grown to mid-log phase in LB and then centrifuged at 4°C to collect the culture supernatant. Secreted proteins were precipitated by addition of 10% trichloroacetic acid (TCA, Sigma-Aldrich) followed by centrifugation for 1 h at 4°C. The pellet was washed twice with acetone and air-dried. Samples were loaded and fractionated under denaturing conditions using SDS-PAGE. Proteins were subsequently stained using Coomassie Brilliant blue R250.

### Western blotting

*Salmonella* cultures were grown in LB for 2 h. Afterwards, cells were washed three times and resuspended in an equal volume of fresh LB. Subsequently, 0.2% arabinose was added and cultures were further incubated for 2 h. Samples were collected before washing and at time points 0, 30, 60 and 120 min after arabinose addition. The samples were centrifuged to separately collect cells and supernatants. Protein precipitation and fractionation from both cells and supernatants were carried out as described earlier. Fractionated proteins were then transferred to a nitrocellulose membrane and proteins were detected using polyclonal (α-FliG, α-FliK, α-FliC,) and monoclonal (α-DnaK) antibodies. α-FliG and α-FliK antibodies were a kind gift from Tohru Minamino (Osaka University). α-FliC and α-DnaK are commercially available from BD Difco and abcam, respectively.

### Transcriptome analysis

Total bacterial RNA was isolated from logarithmically grown cultures by a hot phenol extraction protocol [114]. Residual DNA was digested using the TURBO DNase kit (Ambion) and RNA qualities were assessed using Agilent Technologies 2100 Bioanalyzer. Ribosomal RNA was depleted using Ribo-Zero rRNA Removal kit for bacteria (Epicenter). Library preparation was done using ScriptSeq kit (Epicenter). Sequencing was performed on HiSeq 2500 (Illumina) using TruSeq SBS Kit v3—HS (Illumina) for 50 cycles. Image analysis and base calling were performed using the Illumina pipeline v 1.8.

### RNA isolation and quantitative real-time PCR

Total RNA was isolated from logarithmically grown cultures using RNeasy mini kit (Qiagen) as described previously [115]. Reverse transcription and quantitative real-time PCRs (qRT-PCR) were performed using the SensiFast SYBR No-ROX One Step kit (Bioline) in a Rotor-Gene Q lightcycler (Qiagen). Relative changes in mRNA levels were analysed according to Livak and Schmittgen [116] and normalized against the transcription levels of multiple reference genes according to the method described by Vandesompele et al. [117]. The reference genes *gyrB* and *gmk* were used as previously described [52].

### Statistical analyses

Statistical analyses were performed with GraphPad Prism 9.0 (GraphPad Software, Inc., San Diego, CA) when three or more biological replicates were available. Values of $P < 0.05$ were considered statistically significant.

## Supporting information

**S1 Fig. HilD induction increases SPI-1 effector protein transcription and secretion and affects motility independent of FlhDC.** (A) Fluorescence intensities of a $P_{sicA}$-eGFP fusion were measured in single cells as a reporter for *hilD* expression levels after inducing its expression from *araBAD* locus ($P_{ara}$) (left) and from the native locus under a tetracycline inducible promoter ($P_{tet}$) (right) using arabinose 0.2% and AnTc 100 ng/ml, respectively. Violin plots represent single-cell data from at least 200 analysed cells from one representative experiment. Strains analysed were EM899 ($P_{ara}$-*hilD*), EM900 ($P_{ara}$-FRT), EM228 (WT) and EM12302 ($P_{tet}$-*hilD*). (B) SPI-1 effector protein secretion into the culture supernatant after HilD induction using arabinose 0.2% was analysed via SDS-PAGE and Coomassie Blue staining. Results from one representative experiment are shown. Strains analysed were EM808 (control), TH16339 (*hilD*↑) and EM93 (*hilD*↑ ΔSPI-1). (C) Induction of SPI-1 gene expression in LB-Miller (1% NaCl) upon *hilE* deletion was monitored using a $P_{sicA}$-eGFP transcriptional reporter fusion (left). Fluorescence intensities of $P_{sicA}$-eGFP fusion. Fluorescence intensities of the cultures were measured in a microplate reader and normalised to the measured $OD_{600}$ to give relative fluorescence units (RFU). The data points represent the calculated mean of twelve biological replicates. Error bars represent standard deviation. Strains analysed were EM228 (control), EM12232 (Δ*hilE*) and EM15052 (ΔSPI-1). Growth rates determined as a function of optical density at 600 nm ($OD_{600}$) under the same conditions are shown to the right. The data points represent the calculated mean of six biological replicates. Error bars represent standard deviation. Strains analysed were TH437 (control), EM12177 (Δ*hilE*) and TH16265 (ΔSPI-1). (D) Swimming motility in soft-agar swim plates containing 1% NaCl was monitored at 37˚C for 3.5 h in the following strains: TH437 (control), EM12177 (Δ*hilE*) and TH16265 (ΔSPI-1). Diameters of swimming halos were measured and normalized to the control (left). Representative swimming halos are shown (right). Data from six biological replicates are shown as individual data points. Horizontal bars (red) represent the calculated mean of biological replicates. (E) HilD-induced motility defect is reversed after removing AnTc used for inducing HilD production by washing with fresh medium. Swimming velocities of single cells were analysed via time-lapse microscopy in LB medium. Strains analysed were TH437 (control) and TH17114 (*hilD*↑). (F) Swimming motility in soft-agar swim plates was monitored at 37˚C for 3.5 h using strains mutated for HilD binding site in the *flhDC* promoter. Diameters of swimming halos were measured and normalized to the control (left). Representative swimming halos are shown (right). Biological replicates are shown as individual data points. Horizontal bars (red) represent the calculated mean of biological replicates. Strains analysed were EM808 (control), TH16339 (*hilD*↑), EM930 (*hilA*↑), EM3050 (control, randomized HilD binding site), EM3051 (*hilD*↑, randomized HilD binding site), EM3052 (*hilA*↑, randomized HilD binding site), EM3059 (control, deleted HilD binding site), EM3060 (*hilD*↑, deleted HilD binding site) and EM3061 (*hilA*↑, deleted HilD binding site). *hilD*↑: strain expressing *hilD* under an inducible promoter. *hilA*↑: strain expressing *hilA* under an inducible promoter. AnTc: anhydrotetracycline. a.u.: arbitrary units.
(TIF)

**S2 Fig. Transcriptome analysis upon HilD overexpression.** (A) PCA plot for the control (EM808) and HilD-induced strains (TH16339) in the presence of arabinose as an inducer for HilD overproduction. Within-sample normalisation was done based on the Transcripts Per Million (TPM). The TPM values of genes less than one variance were removed. The log2 of the TPM values were used to make the PCA plot. (B) Volcano plot showing all differentially expressed genes. DESeq2 was used to calculate the log2(fold change) and the corresponding adjusted *p*-values for the genes, by comparing the expression profile of control samples against

HilD-induced samples. The results in (A) and (B) are calculated from two biological replicates. *hilD*↑: strain expressing *hilD* under an inducible promoter.
(TIF)

**S3 Fig. Growth rates of strains overproducing different adhesins in *trans*.** Growth rates were determined as a function of optical density at 600 nm (OD600) measured in a plate reader in absence (left) and presence (right) of 100 ng/ml AnTc as an inducer. The data points represent the calculated mean of six biological replicates at each time point. Error bars represent standard deviation. Strains analysed were EM12144 (VC, vector control), EM12145 (p*csg*), EM12146 (p*saf*), EM12147 (p*std*), EM12148 (p*pef*). AnTc: anhydrotetracycline.
(TIF)

**S4 Fig. HilD-activation results in a growth defect.** (A) Growth rates determined as a function of optical density at 600 nm ($OD_{600}$) in absence and presence of AnTc for inducing HilD production. Individual points represent means of six biological replicates at each time point. Error bars represent the standard deviation. *hilD*↑: strain expressing *hilD* under an inducible promoter. AnTc: anhydrotetracycline. Strains analysed were TH437 (control), TH17114 (*hilD*↑), EM12479 (*hilD*↑ ΔSPI-1). (B) HilD induction results in decreased cell length (left) and coccoid morphology (right). Cell length (μm) of individual bacteria was determined by phase-contrast microscopy. Individual data points represent the averages of the single-cell lengths of independent experiments. Violin plots represent all single cell data values from a total of at least 400 analysed single cells. Horizontal bars (bold) represent the calculated average of three independent experiments. The error bars represent the standard error of mean and statistical significances were determined using a two-tailed Student's t-test (***, $P < 0.001$; ns, $P > 0.05$). Representative images are shown in the right panel. the same strains as in (A) were analysed. (C) Time-lapse microscopy analysis reveals a decreased cell elongation rate (left), a decrease in the maximal cell length (middle) as well as decreased length elongation during a division cycle (right) upon HilD induction. Violin plots represent data values from at least 300 analysed cell lineages from a representative experiment. Strains analysed were EM12802 (control) and EM12803 (*hilD*↑).
(TIF)

**S5 Fig. *rpoS*-mCherry functions as a reporter for the stringent response.** (A) Fluorescence intensities of *rpoS*-mCherry C-terminal translation fusions were measured in a strain overexpressing a constitutively active RelA mutant comprised of the first N-terminal 455 amino acids (*relA*↑). Fluorescence intensities in single cells were quantified using fluorescence microscopy (left). Violin plots represent data values from at least 200 analysed single cells from a representative experiment. Representative images are shown (right). Strains analysed were EM13226 (VC) and EM13227 (*relA*↑). (B) Assessment of *rpoS*-mCherry response to nutrient limitation in M9 minimal medium. Fluorescence intensities in single cells were quantified using fluorescence microscopy (left). Violin plots represent data values from at least 200 analysed single cells from a representative experiment. Representative images are shown (right). Scale bar is 2 μm. Strain analysed was EM13017. a.u.: arbitrary units.
(TIF)

**S6 Fig. *rflP*-mScarlet expression under HilD inducing conditions.** Strains were grown in presence of AnTc to induce HilD overproduction. Fluorescence intensities of *rflP*-mScarlet translational fusion were quantified in single-cells (left). Representative microscopy images are shown (right). Scale bar is 2 μm. Individual data points represent the averages of the single-cell data of independent experiments. Violin plots represent data values from at least 700 analysed single cells. Horizontal bars (bold) represent the calculated average of three independent

experiments. Error bars represent standard error of the mean. Statistical significances were determined using a two-tailed Student's t-test (ns, $P > 0.05$). Strains analysed were EM13097 (control), EM13278 (*hilD*↑) and EM13363. (*hilD*↑ ΔSPI-1). *hilD*↑: strain expressing *hilD* under an inducible promoter. AnTc: anhydrotetracycline. a.u.: arbitrary units.
(TIF)

**S7 Fig. DiSC$_3$(5) responds to CCCP-induced membrane depolarization.** Strains were grown under HilD inducing conditions using AnTc followed by treatment with CCCP. Cells were then stained with the membrane potential sensitive dye DiSC$_3$(5) and single-cell fluorescence intensities were quantified (left). Violin plots represent data values from at least 100 analysed single cells from a representative experiment. Representative microscopy images are shown (right). Scale bar is 2 μm. Strains analysed were TH437 (control), TH17114 (*hilD*↑) and EM12479. (*hilD*↑ ΔSPI-1). *hilD*↑: strain expressing *hilD* under an inducible promoter. AnTc: anhydrotetracycline. a.u.: arbitrary units.
(TIF)

**S8 Fig. Expression of flagella proteins upon HilD-induction.** The levels of FliG, FliK and FliC were determined using western blotting in either the cellular fraction or the supernatant of the bacterial cultures as indicated. DnaK was used as a loading control. Samples were collected before HilD-induction and at the indicated time points after washing and inducing HilD by the addition of 0.2% arabinose. Results from a representative experiment are shown. Strains analysed were EM900 (control) and EM899 (*hilD*↑).
(TIF)

**S1 Table.** *Salmonella enterica* serovar Typhimurium strains used in this study.
(DOCX)

**S2 Table. Plasmids used in this study.**
(DOCX)

**S3 Table. Primers used in this study.**
(DOCX)

**S4 Table. Transcriptional profile upon HilD induction in transcript per million (TPM).**
(XLSX)

## Acknowledgments

We thank Michael Hensel (Universität Osnabrück) and Tohru Minamino (Osaka University) for kindly providing plasmids and antibodies, respectively. We are thankful to Philipp F. Popp for his help with establishing single cell assays. We also thank Heidi Landmesser and Raúl Trepel for expert technical assistance. We are grateful for the Erhardt lab members for their valuable comments on the manuscript.

## Author Contributions

**Conceptualization:** Doaa Osama Saleh, Julia A. Horstmann, Marc Erhardt.

**Data curation:** Abilash Chakravarthy Durairaj, Till Strowig.

**Formal analysis:** Doaa Osama Saleh, Julia A. Horstmann, Willi Weber, Abilash Chakravarthy Durairaj.

**Funding acquisition:** Till Strowig, Marc Erhardt.

**Investigation:** Doaa Osama Saleh, Julia A. Horstmann, María Giralt-Zúñiga, Willi Weber, Abilash Chakravarthy Durairaj.

**Methodology:** Doaa Osama Saleh, Julia A. Horstmann, María Giralt-Zúñiga, Willi Weber, Eugen Kaganovitch, Abilash Chakravarthy Durairaj.

**Project administration:** Marc Erhardt.

**Resources:** Marc Erhardt.

**Supervision:** Enrico Klotzsch, Till Strowig, Marc Erhardt.

**Validation:** Doaa Osama Saleh, Julia A. Horstmann, María Giralt-Zúñiga.

**Visualization:** Doaa Osama Saleh, Julia A. Horstmann, Willi Weber, Enrico Klotzsch, Marc Erhardt.

**Writing – original draft:** Doaa Osama Saleh, Julia A. Horstmann, María Giralt-Zúñiga, Willi Weber.

**Writing – review & editing:** Doaa Osama Saleh, Marc Erhardt.

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
