## [Decision Letter · Decision Letter 0]

20 Mar 2023

Dear Ms. Saleh,

Thank you very much for submitting your manuscript "SPI-1 virulence gene expression modulates motility of Salmonella Typhimurium in a proton motive force- and adhesins-dependent manner" for consideration at PLOS Pathogens. As with all papers reviewed by the journal, your manuscript was reviewed by members of the editorial board and by several independent reviewers. In light of the reviews (below this email), we would like to invite the resubmission of a significantly-revised version that takes into account the reviewers' comments.

We cannot make any decision about publication until we have seen the revised manuscript and your response to the reviewers' comments. Your revised manuscript is also likely to be sent to reviewers for further evaluation.

Sincerely,

Sophie Helaine

Academic Editor

PLOS Pathogens

Nina Salama

Section Editor

PLOS Pathogens

Kasturi Haldar

Editor-in-Chief

PLOS Pathogens

orcid.org/0000-0001-5065-158X

Michael Malim

Editor-in-Chief

PLOS Pathogens

orcid.org/0000-0002-7699-2064

Reviewer's Responses to Questions

**Part I - Summary**

Reviewer #1: The manuscript by Saleh et al explores the effects of HilD activation on motility, stringent response and adhesin production in Salmonella Typhimurium. Specifically, they show that despite its role in activation of flhD expression, HilD overexpression reduces motility and increases the expression of several previously described adhesin gene clusters. The authors further tie this phenotype to SPI-1-dependent induction of the stringent response and decrease of PMF. The experiments are elegant, the data is convincing and the setups are well-described. Furthermore, the work touches upon an interesting topic of how certain transcriptional programs in S. Typhimurium can dissipate the cellular energy pool, ultimately leading to phenotypic changes. I have only spotted a few points that the authors might want to consider prior to publication.

Reviewer #2: Saleh et al. conclude that induction of the SPI1 T3SS leads to a loss of PMF and hence, loss of motility. Various adhesins are also induced. The overall data support the conclusion. This leads to an intriguing model in which the cell turns on SPI1 and stops swimming, potentially enhancing invasion. The issue is that this model is counter to the coordinate transcriptional induction of the two systems, a point that needs to be more fully discussed.

Reviewer #3: In this manuscript, Saleh and colleagues describe the effect of overexpression of HilD, the master transcriptional activator of the Salmonella SPI-1 injectisome, on bacterial cell physiology, especially motility. This study follows up on an earlier study by the same group (Singer et al., J Bact 2014), where HilD overexpression led to increased transcription of flhDC, the master transcription activator of the flagellum. In apparent contrast, the current study shows a strong reduction of motility upon HilD overexpression, which depends on the presence of SPI-1. Overexpression of HilD in presence of SPI-1 also resulted in an activation of the stringent response and reduced membrane potential. The authors propose that these data indicate a strategy to quickly change the bacterial lifestyle from mobile to adhesive upon contact to host cells.

The interplay of different molecular mechanisms (such as motility and protein secretion) is an important and so far poorly understood aspect of bacterial virulence. This manuscript addresses it with a wide range of experiments, which are presented in a clear and very attractive manner. Similarly, the proposed model is intuitive. However, the data mainly describe effects of overexpression of a known transcription factor, HilD, and confirm that these effects depend on the presence of its known main target, the SPI-1 injectisome. Whether and to which extent the results show a biologically relevant mechanism therefore remains largely unclear. Further, the reason for the – at least on the first look – clear discrepancy between the results of the 2014 paper and this manuscript is not revealed.

**Part II – Major Issues: Key Experiments Required for Acceptance**

Reviewer #1: 1. Although overexpression of HilD results in a set of well described and convincing phenotypes, I wonder if the authors may want to discuss how these phenotypes might change, if experiments had been performed under more physiologically relevant conditions. Using LB with 0.3M NaCl might be one option. It should be discussed if that accumulation of HilD and concomitant overexpression of type 3 secretion systems in the presented experiments might result not only affect the normal hilD regulon, but also yield some unspecific response phenotypes (Fig. S2B, differential expression of ca 1600 genes doesn’t look hilD-specific anymore). Could this be approached by comparing the published transcriptome data of hilD mutants vs wt (Colgan et al) side by side with the new data on inducible hilD expression?

2. The authors cite literature showing the roles of PMF not only in flagellar rotation, but also in flagellar synthesis (lines 334-335). They additionally link the induction of stringent response upon HilD overexpression to PMF dissipation (Figure 6). However, I am not sure why the flagellation level of such cells remained largely unaffected (Figure 5). Can the authors comment on that?

3. Figure 7: the model proposed by the authors relies on the assumption that HilD is only activated (resulting in repressed motility) upon the contact between STm and epithelial cells. However, it has been previously shown that HilD is activated by several environmental cues in the gut lumen (Ellermeier et al, 2008; Golubeva et al, 2012; Diard et al, 2013). I would say that the lines 419-430 are indeed highly speculative and not backed up by corresponding data. The authors may want to modify this statement.

Reviewer #2: None

Reviewer #3: 1. Almost all experiments were performed with strains overexpressing HilD (usually in wild-type and SPI-1 deletion backgrounds). A direct comparison of wild type and SPI-1 deletion without the overexpression of HilD would often have been possible and more physiologically relevant. Although overexpression of a transcriptional activator can yield insights into its working mechanism, interpretation of physiological effects of native expression of the target genes are difficult. The fact that strong overexpression of HilD and in turn the SPI-1 injectisome (a large membrane-spanning complex), leads to effects including loss of membrane potential, does not necessarily allow conclusions about the native interaction of SPI-1 expression and motility, as claimed in the title and throughout the manuscript. These claims should therefore be corrected.

2. The reason for the apparent contradiction between the results of Singer et al. (overexpression of HilD leads to increased transcription of flhDC) and this manuscript is a critical question, but remains unclear. The manuscript almost actively steers clear from data that would allow to clarify this point. For example, the transcriptomics data shown in Figure 2A for “selected genes” only includes two flagellar genes. To evaluate the data and any differences between the two studies (and potential reasons), the complete dataset must be made available, which can be done using a preliminary protected link for the reviewers. In addition, at least the genes presented in Figure 2A plus all flagellar genes must be listed with their names and the actual data in a Supplementary Table. Furthermore, it would be very helpful to show the protein levels of at least some flagellar components.

3. The abstract states that “overproduction of several adhesion systems phenocopied the HilD-induced motility defect”. While this is true for swimming velocities in a glass flow chamber (which, as the authors themselves state, is probably an artifact of attachment to the glass surface), this does not seem to be the case for the other observed phenotypes. Relatedly, the following sentence states that “combination of SPI-1-dependent depletion of the PMF and upregulation of adhesins (…) allows flagellated Salmonella to rapidly modulate their motility.” Given that deletion of SPI-1 alone rescues most phenotypes, it appears to be the main (or only) factor for them and the role of adhesins minor, if present at all.

4. Is it really that surprising to find that flagella and SPI-1 injectisomes are adversely regulated in Salmonella? Similar mutually negative regulation has been found in several other cases, including E. coli, Yersinia enterocolitica and Pseudomonas aeruginosa (Iyoda et al, J Bact (2006); Bleves et al, J Bact (2002); Soscia et al. J Bact (2007)). These references should be discussed and the text adapted accordingly.

**Part III – Minor Issues: Editorial and Data Presentation Modifications**

Reviewer #1: 1. Line 268: It would be interesting to compare the growth rate and cell length phenotypes of hilD expression to the published data from Arnoldini et al 2014 and Hockenberry et al. 2021.

2. Lines 312-320: Authors rule out the potential role of RflP in the observed motility phenotype. However, in contrast to the quantification data, the microscopy image suggests a difference. The authors may want to choose a more representative image?

3. Line 56: no need for comma after “although”

4. Line 149: “exclude”, not “exclude for”

5. Statistical test results should be added in Fig. 1, Fig. 2BD, Fig. 3A

6. Fig. 2DE. It would be interesting for the readers, if the authors could discuss the differences in the two motility phenotypes of pcsg, psaf and pstd.

7. Line 337-343: The authors may want to discuss that besides T3SS effector secretion, errors or normal steps of T3SS apparatus assembly may also dissipate the PMF.

8. References 65 and 73 appear to be partially deleted?

Reviewer #2: 1. Line 79. “A battery of DNA-binding proteins, such as … HilD, HilC, and RtsA…” While numerous regulatory systems affect expression of SPI1, most function indirectly. This sentence makes HilD seem like a minor player.

2. Line 82. A pet peeve. I think calling HilA and an “OmpR/ToxR-type transcriptional regulator” can be confusing. The uninitiated might think it is a two-component regulator. It is only the DNA binding domain that is similar to OmpR.

3. The regulatory interplay between flagella and SPI1 is obviously complicated. You could use a model slide to make it clear.

4. Line 137. The strain description is not clear. “…a chromosomal copy of hilD under the control of an inducible promoter, including an ectopic locus (araBAD) and the native locus.” Is the “native locus” a tet promoter? Clarify. You also should point out that the native locus is intact in the ara:hilD strain.

5. Line 145. You do not describe the strain ectopically expressing hilA.

6. Fig 2B. You need statistical analyses. Is any of this significant?

7. Line 195. “Strains expressing hilD under the native promoter and deleted for SPI1…” This makes no sense. If SPI1 is deleted, so is hilD. This strain is ara::hilD, correct?

8. Line 231. Deletion of SPI1 blocks the effect. In your strains, deletion of SPI1 is likely to significantly decrease the overall amount of HilD/HilC/RtsA in the cell, given the ectopic HilD expression will induce the native loci with autoinduction intact. Your data seem clear and induction of hilA also confers the phenotype. However, you should likely acknowledge this caveat.

9. Line 264. What is the evidence that “translation rate” per se is affected? The word “translation” is not used in Ref 53.

10. Figure 6. How long were these cells induced? Does time matter?

11. Although your model is intriguing, it goes against the apparent coordinate transcriptional induction of the two systems. Why would HilD turn on FlhDC if the goal is to stop swimming? You need to at least comment on this. I am wondering if the timing of these events matters and perhaps needs more careful explanation and consideration. The materials and methods implies that the cells in Fig 6 were incubated for 5 mins? Fig 1 suggests that loss of motility occurs after 45 mins?

Reviewer #3: 1. The number of biological replicates should be indicated for each experiment.

2. In the proposed model, would the reversibility of the commitment to attachment (e.g., in case of failed interactions with the host cell) be an advantage of the mechanism proposed in the authors’ model over the ejection of flagella?

3. Line 91, ref. 29 does not deal with injectisomes, please add suitable reference.

4. Line 98, please describe how FliZ regulates HilD in ref. 34.

5. Line 110, in contrast to the claim that “HilD activates the expression of SPI-2 genes”, ref. 43 explicitly states (in the abstract) that “HilD is not required for SPI-2 regulon expression under the in vitro growth conditions that are thought to resemble the intracellular environment”.

PLOS authors have the option to publish the peer review history of their article (what does this mean?). If published, this will include your full peer review and any attached files.

Reviewer #1: No

Reviewer #2: No

Reviewer #3: No

Figure Files:

Data Requirements:

Please note that, as a condition of publication, PLOS' data policy requires that you make available all data used to draw the conclusions outlined in your manuscript. Data must be deposited in an appropriate repository, included within the body of the manuscript, or uploaded as supporting information. This includes all numerical values that were used to generate graphs, histograms etc.. For an example see here on PLOS Biology: http://www.plosbiology.org/article/info:doi%2F10.1371%2Fjournal.pbio.1001908#s5.
---

## [Editor Report · Decision Letter 1]

1 Jun 2023

Dear Ms. Saleh,

We are pleased to inform you that your manuscript 'SPI-1 virulence gene expression modulates motility of Salmonella Typhimurium in a proton motive force- and adhesins-dependent manner' has been provisionally accepted for publication in PLOS Pathogens.

Best regards,

Sophie Helaine

Academic Editor

PLOS Pathogens

Nina Salama

Section Editor

PLOS Pathogens

Kasturi Haldar

Editor-in-Chief

PLOS Pathogens

orcid.org/0000-0001-5065-158X

Michael Malim

Editor-in-Chief

PLOS Pathogens

orcid.org/0000-0002-7699-2064
---

## [Editor Report · Acceptance letter]

12 Jun 2023

Dear Ms. Saleh,

We are delighted to inform you that your manuscript, "SPI-1 virulence gene expression modulates motility of *Salmonella* Typhimurium in a proton motive force- and adhesins-dependent manner," has been formally accepted for publication in PLOS Pathogens.

Best regards,

Kasturi Haldar

Editor-in-Chief

PLOS Pathogens

orcid.org/0000-0001-5065-158X

Michael Malim

Editor-in-Chief

PLOS Pathogens

orcid.org/0000-0002-7699-2064